# Essential Amino Acid Starvation-Induced Oxidative Stress Causes DNA Damage and Apoptosis in Murine Osteoblast-like Cells

**DOI:** 10.3390/ijms242015314

**Published:** 2023-10-18

**Authors:** Runbo Li, Hirohito Kato, Chihiro Fumimoto, Yurika Nakamura, Kimihiro Yoshimura, Emika Minagawa, Keiju Omatsu, Chizuko Ogata, Yoichiro Taguchi, Makoto Umeda

**Affiliations:** Department of Periodontology, Osaka Dental University, Hirakata, Osaka 573-1121, Japan; li-r@cc.osaka-dent.ac.jp (R.L.); kato-h@cc.osaka-dent.ac.jp (H.K.); fumimoto-c@cc.osaka-dent.ac.jp (C.F.); nakamura-y@cc.osaka-dent.ac.jp (Y.N.); yoshimura-k@cc.osaka-dent.ac.jp (K.Y.); minagawa-e@cc.osaka-dent.ac.jp (E.M.); omatsu-k@cc.osaka-dent.ac.jp (K.O.); ogata-c@cc.osaka-dent.ac.jp (C.O.); umeda-m@cc.osaka-dent.ac.jp (M.U.)

**Keywords:** essential amino acids, apoptosis, ROS, osteoblasts

## Abstract

Intracellular nutrient metabolism, particularly the metabolism of essential amino acids (EAAs), is crucial for cellular functions, including energy production and redox homeostasis. An EAA deficiency can lead to cellular dysfunction and oxidative stress. This study explores the mechanisms underlying cellular responses to EAA starvation, focusing on ROS-induced DNA damage and apoptosis. MC3T3-E1 cells were subjected to EAA starvation, and various assays were conducted to assess cell proliferation, survival, DNA damage, and apoptosis. The antioxidant N-acetylcysteine (NAC) was employed to block ROS formation and mitigate cellular damage. Gene expression and Western blot analyses were performed to elucidate molecular pathways. EAA starvation-induced ROS generation, DNA damage, and apoptosis in MC3T3-E1 cells. NAC administration effectively reduced DNA damage and apoptosis, highlighting the pivotal role of ROS in mediating these cellular responses during EAA deficiency. This study demonstrates that EAA starvation triggers ROS-mediated DNA damage and apoptosis, offering insights into the intricate interplay between nutrient deficiency, oxidative stress, and programmed cell death. NAC emerges as a potential therapeutic intervention to counteract these adverse effects.

## 1. Introduction

Intracellular nutrient metabolism involves intricate biochemical pathways and enzymatic reactions that enable cells to obtain, store, and utilize nutrients for energy production, macromolecule biosynthesis, cellular signaling regulation, and redox homeostasis [1]. One crucial intracellular nutrient metabolism is that of amino acids; it is crucial for cell survival and function and is regulated by a complex interplay of cellular and molecular mechanisms [2].

Protein synthesis is the primary pathway involved in essential amino acid (EAA) metabolism. There are nine EAAs (histidine, leucine, lysine, methionine, phenylalanine, isoleucine, threonine, tryptophan, and valine), and cells utilize them to synthesize proteins that are essential for cellular structure, function, and regulation [3]. Protein synthesis occurs in cellular organelles called ribosomes. During protein synthesis, amino acids are linked via peptide bonds in a specific order according to the genetic code carried by the messenger RNA (mRNA) [4]. The availability of EAAs in the cell is critical for proper protein synthesis because deficiencies can lead to impaired cellular function and overall health [5].

Moreover, intracellular amino acid metabolism plays a critical role in maintaining cellular redox balance and protecting cells from oxidative damage caused by reactive oxygen species (ROS) [6]. ROS are oxygen-containing, chemically reactive molecules that are generated as natural byproducts of cellular metabolism [7]. They play important roles in various cellular processes, including cell signaling and the regulation of cellular homeostasis [8]. At low-to-moderate levels, ROS can act as secondary messengers in signaling pathways and participate in essential cellular functions [9,10]. However, when ROS levels exceed the antioxidant defense capacity of these cells, oxidative stress is induced [11]. Oxidative stress can damage proteins, lipids, and the DNA within cells, leading to cellular dysfunction and potentially even apoptosis [12].

Apoptosis, often referred to as programmed cell death, is a natural process that regulates the cell population during development and tissue homeostasis [13]. It is a controlled mechanism that helps remove damaged, infected, or unnecessary cells from the body. Apoptosis involves a series of well-orchestrated biochemical events that culminate in the dismantling and removal of cells without releasing toxic or inflammatory contents into the surrounding tissues [14].

We previously examined cellular changes brought on by EAA deprivation and studied the inhibition of osteogenic differentiation in murine osteoblasts [15]. However, the precise effects of EAA deprivation on the interaction between oxidative stress and apoptosis remain unclear. The primary objective of this study, therefore, was to investigate the cellular alterations that occur in the absence of EAAs and the underlying mechanisms. In this study, we employed N-Acetylcysteine (NAC), a versatile drug and supplement with various clinical applications, to eliminate the effects of ROS. NAC’s uses range from serving as an antidote to supplementing cellular glutathione oxidants and treating specific psychiatric disorders. The findings of this study are anticipated to significantly advance our understanding of how cells respond to EAA deficiencies, seeking to bridge the existing knowledge and gap surrounding cellular responses to EAA deficiency and gain insights into the molecular pathways implicated in these cellular changes. The findings of this study are expected to considerably advance our understanding of how cells respond to EAA deficiencies and provide valuable information regarding the potential molecular targets involved in this process.

## 2. Results

### 2.1. EAA Starvation Induces Oxidative Stress in MC3T3-E1 Cells

ROS production in MC3T3-E1 cells subjected to EAA starvation (−EAA) was evaluated after 24, 72, and 120 h of incubation. As shown in Figure 1, the ROS level induced in the −EAA group was significantly higher than that in the control group (+EAA).

The redox status of cells was assessed using gluthione (GSH) and oxidized glutathione (GSSG) production levels. In the −EAA group, both GSSG production and the GSSG/GSH ratio significantly increased.

### 2.2. EAA Starvation Impairs Cell Viability in MC3T3-E1 Cells

MC3T3-E1 cell proliferation under EAA starvation was assessed after 24, 72, and 120 h of incubation. As shown in Figure 2, DNA replication (BrdU incorporation) was significantly inhibited in the −EAA group. Moreover, DNA content in MC3T3-E1 cells subjected to EAA starvation was notably lower than that in the control group. This indicated a reduction in cell proliferation due to EAA deprivation. Furthermore, the survival rate of MC3T3-E1 cells was significantly reduced in the EAA-deprived environment.

Additionally, a significant increase in the levels of γH2AX, a marker of DNA damage, was observed after 24 h of EAA starvation.

### 2.3. EAA Starvation Induces Apoptosis in MC3T3-E1 Cells

As depicted in Figure 3, the apoptosis rate under the EAA starvation condition significantly increased after 24 h of incubation compared with that in the control group.

Moreover, the gene expression levels of p53 and Bax increased significantly under EAA starvation after 24, 72, and 120 h of incubation, and the protein expression levels of p53, Bax, and cleaved caspase-3 increased significantly after 24, 72, and 120 h of incubation.

### 2.4. NAC Administration Inhibits ROS Production

The effect of NAC on ROS production under glucose starvation was evaluated after culturing cells for 24, 72, and 120 h. As shown in Figure 4, ROS production was significantly inhibited in all NAC-treated groups (1, 2.5, and 5 mM). The most substantial inhibition was observed at a NAC concentration of 5 mM.

Furthermore, in the NAC treatment group, both the GSSG production and the GSSG/GSH ratio were significantly decreased compared with those in the EAA starvation group.

### 2.5. NAC Administration Rescues Cell Viability under EAA Starvation

MC3T3-E1 cell proliferation and survival were also measured under EAA starvation conditions. As shown in Figure 5, a significant rescue of cell proliferation and survival by an NAC concentration of 5 mM was observed.

In addition, NAC administration significantly inhibited γH2AX expression under EAA starvation after 24 h of incubation.

### 2.6. NAC Application Inhibits Apoptosis and Apoptosis-Related Gene Expression in MC3T3-E1 Cells under EAA Starvation

Figure 6 presents the effects of NAC application under EAA starvation on apoptosis in MC3T3-E1 cells, which were determined using FACS and immunofluorescence staining. A significant reduction in the apoptotic rate of MC3T3-E1 cells in the NAC-treated groups was observed after 24 h of incubation. 

Furthermore, NAC administration significantly reduced the EAA starvation-induced increase in p53 and Bax expression at all time points. Moreover, the protein expression levels of p53, Bax, and cleaved caspase-3 were significantly decreased after 24, 72, and 120 h of incubation in the NAC-treated group compared with those in the −EAA group.

## 3. Discussion

Previously, we made significant discoveries on the impact of specific environmental conditions on cellular behavior [16,17,18]. In one study, we demonstrated the induction of ROS formation under low-glucose conditions, leading to cellular damage. In another study, we found that EAA deprivation resulted in cell cycle arrest, primarily mediated through the phosphorylation of the MAPK signaling pathway. This, in turn, inhibited both cell proliferation and osteogenic differentiation [15]. 

MC3T3-E1 is a mouse osteoblast cell line commonly used in research related to bone biology and osteogenesis, which are frequently utilized in vitro to study various aspects of bone biology, including osteoblast differentiation, mineralization, and responses to different factors. In the present study, we examined the effects of EAA starvation on ROS formation. Similar to glucose starvation, EAA deprivation triggered ROS generation in MC3T3-E1 cells. To gain a deeper understanding of the cellular redox environment, the ratio of oxidized glutathione (GSSG) to reduced glutathione (GSH) in the cells was measured. The balance between GSSG and GSH is crucial for cellular redox homeostasis [19]. An increase in ROS levels can lead to oxidative stress, increasing GSSG levels and altering the GSSG-to-GSH ratio. Through assessing this ratio, we found that MC3T3-E1 cells were in a state of high oxidative stress under EAA starvation.

Among the EAAs, methionine plays a unique and important role because of its high sensitivity to various forms of ROS and its natural antioxidant properties [20]. Additionally, methionine serves as a precursor to several crucial molecules such as S-adenosylmethionine, hydrogen sulfide, taurine, and glutathione, all of which effectively attenuate the oxidative stress induced by different oxidants and protect tissues from damage [21,22]. In the context of EAA starvation-induced oxidative stress, methionine deficiency may play a significant role in contributing to this heightened state of oxidative stress. Methionine depletion can disrupt the delicate balance between ROS production and scavenging mechanisms, potentially exacerbating cellular oxidative damage.

Subsequently, we evaluated cell viability by examining their proliferative activity and survival rates. A noteworthy reduction in both the proportion of BrdU bound to DNA and the overall amount of DNA in the EAA-starved group was found. These findings strongly suggest that the proliferative capacity of MC3T3-E1 cells was significantly impaired upon EAA starvation. In the EAA-starved group, we observed a significant decrease in cell survival rate and a concurrent increase in DNA damage. Previous studies have established that elevated ROS levels can impair the function and viability of various cell types [16]. Our study provides additional evidence that EAA starvation-induced ROS generation could impair cell viability.

ROS can modulate apoptotic signaling in a nuanced manner. Within certain physiological ranges, ROS can act as secondary messengers in various signaling pathways, such as those of NF-κB and AP-1, which regulate cell survival and death [23,24]. Additionally, ROS can activate pro-apoptotic proteins, such as p53 and Bax, while inhibiting anti-apoptotic proteins, such as Bcl-2, thereby tipping the scales in favor of apoptosis [25,26]. High ROS levels directly damage cellular DNA, leading to DNA strand breaks and mutations. If DNA damage is severe and beyond repair, cells can activate DNA damage response pathways that induce apoptosis to prevent the proliferation of potentially dangerous and genetically unstable cells [12]. In this study, we confirmed a significant increase in the apoptosis rate after EAA starvation, with the increased expression of pro-apoptotic genes and proteins, p53 and Bax. Meanwhile, the level of the apoptotic protein cleaved caspase-3 also increased.

These findings confirm the role of EAA starvation-induced ROS in triggering apoptosis via alterations in the expression of key apoptotic signaling proteins. Understanding the intricate interplay between ROS and apoptosis can contribute to our understanding of the cellular responses to nutrient deprivation and oxidative stress, offering potential insights into therapeutic approaches for apoptosis-associated disorders.

To gain deeper insights into the role of ROS under EAA starvation, we used NAC, an antioxidant widely acknowledged for its free-radical scavenging activity, particularly oxygen radicals. NAC possesses potent antioxidant properties and is recognized as a potential therapeutic alternative for various disorders caused by free oxygen radical production [27]. In the present study, NAC was used to inhibit ROS formation during EAA starvation. Following NAC treatment, cellular viability, proliferative activity, and survival rate showed remarkable increases. This suggests that the reduced cellular viability observed during EAA starvation could be attributed to ROS-induced oxidative damage. Moreover, DNA damage was significantly decreased after NAC treatment, suggesting that the detrimental effects of ROS in EAA-starved cells can be effectively mitigated by NAC.

Next, we investigated the effect of NAC treatment on apoptosis in MC3T3-E1 cells under EAA starvation. Some studies have shown that NAC prevents apoptosis [28,29]. In this study, NAC treatment resulted in a notable reduction in the apoptosis rate and significantly decreased the expression of the key pro-apoptotic proteins p53, Bax, and cleaved caspase-3. These results indicate that the antioxidant properties of NAC effectively mitigate the upregulation of pro-apoptotic factors induced by EAA starvation.

In the present study, we confirmed that EAA starvation induces ROS generation, DNA damage, and apoptosis in MC3T3-E1 cells. Remarkably, NAC administration decreased both DNA damage and apoptosis. This observation demonstrates that ROS generated during EAA starvation is responsible for inducing DNA damage and subsequent apoptosis in MC3T3-E1 cells. Furthermore, the ability of NAC to rescue cellular damage induced by ROS highlights its potential therapeutic significance in countering apoptosis under EAA deficiency. Moreover, the efficacy of NAC in ameliorating cellular damage induced by ROS underscores its potential therapeutic significance in mitigating apoptosis resulting from EAA deficiency. This discovery not only highlights the central role of ROS in orchestrating observed cellular responses but also positions NAC as a promising candidate for therapeutic interventions to counteract the deleterious effects of apoptosis in the context of EAA deficiency. By specifically targeting ROS-mediated DNA damage and apoptosis, NAC emerges as a compelling candidate for therapeutic strategies that are aimed at mitigating the adverse cellular effects associated with EAA deficiency. This has far-reaching implications for the development of targeted interventions to enhance cellular resilience and overall clinical outcomes in situations of EAA deficiency.

Studies have demonstrated a direct correlation between the induction of apoptosis and the phosphorylation of c-Jun N-terminal kinase (JNK) and p38 mitogen-activated protein kinase (p38) [30]. JNKs initiate apoptotic signaling through two mechanisms: first, by increasing the expression of pro-apoptotic genes through the activation of specific transcription factors and second, by directly modulating mitochondrial pro- and anti-apoptotic proteins through distinct phosphorylation events [31]. Similarly, p38 regulates apoptosis by phosphorylating Bim (EL) Ser-65 and also induces apoptosis by phosphorylating Bcl-2 family members [32]. We previously established that EAA starvation induces p38 and JNK phosphorylation [15]. This indicates that the apoptosis observed in the current study could be related to the phosphorylation of these MAP kinases.

The findings of this study contribute significantly to our understanding of the interplay among ROS, DNA damage, and apoptosis during EAA starvation. However, a major limitation of this study is that we did not examine the individual roles of each EAA when mediating oxidative stress and apoptosis. Furthermore, this study did not explore other potential oxidative responses resulting from the deprivation of EAAs, such as the effect on Malondialdehyde (MDA) in lipid peroxidation reactions. Investigating these additional facets of oxidative stress could offer a more comprehensive understanding of the broader effects induced by EAA starvation. Recognizing these limitations, future research endeavors may benefit from a more detailed exploration of the individual roles of specific EAAs and an expanded examination of oxidative responses beyond the scope of DNA damage and apoptosis. A more in-depth investigation of the specific molecular mechanisms underlying our observations could be highly valuable for unraveling the intricate signaling pathways and cellular responses that occur during EAA starvation. This could yield valuable insights into oxidative stress responses and programmed cell death under various physiological and pathological conditions. Moreover, replicating this experiment in an animal model poses challenges due to the absence of dedicated transporters for each essential amino acid. Blocking the uptake of a specific amino acid becomes intricate since essential amino acids share various transport systems rather than demonstrating individualized transporters. To address this complexity, our future studies aim to attain a similar effect by selectively inhibiting these shared transporters in in vivo experiments.

## 4. Materials and Methods

### 4.1. Cell Culture Media

MC3T3-E1 cells, an immortalized cell line, were purchased from RIKEN BioResource Research Center (Tsukuba, Ibaraki, Japan) and cultured in Dulbecco’s Modified Eagle’s Medium (Nacalai Tesque, Kyoto, Japan) with or without the nine EAAs and supplemented with 10% fetal bovine serum (Hyclone, Thermo Fisher Scientific, Waltham, MA, USA).

### 4.2. Antibodies and Reagents

All antibodies and reagents were purchased from the following sources. Antibiotics (penicillin, streptomycin, and amphotericin B) were bought from Nacalai Tesque. N-Acetyl-L-cysteine (NAC, A7250) was purchased from Sigma-Aldrich (Burlington, MA, USA). Antibodies against p53, Bax, caspase-3, cleaved caspase-3, and β-actin were acquired from Cell Signaling Technology (Beverly, MA, USA).

### 4.3. ROS Staining and Detection

ROS levels were assessed under EAA starvation conditions using the Total ROS Detection Kit obtained from Dojindo Laboratories (Kumamoto, Japan). MC3T3-E1 cells were cultured and allowed to adhere for 24 h, and the medium was changed to one with or without EAAs. The stained cells were observed and photographed using a BZ-X800 all-in-one fluorescence microscope (Keyence, Osaka, Japan) after incubation for 24, 72, and 120 h. Cellular fluorescence was measured using version 7.0 SoftMax Pro Microplate Data Acquisition and Analysis software (Molecular Devices, Sunnyvale, CA, USA).

### 4.4. Glutathione Quantification Assay

The glutathione quantification was performed using the GSSG/GSH Quantification Kit (G257, Dojindo). Briefly, MC3T3-E1 cells were lysed with a lysis buffer (5% of 5-sulfosalicylic acid dihydrate, Nacalai Tesque) and centrifuged for 10 min at 4 °C and 8000× *g*. The absorbance of the supernatant was measured at a 405 nm wavelength and analyzed using = SoftMax^®^ Pro Microplate Data Acquisition and Analysis software.

### 4.5. Detection of Cell Proliferation

Cell proliferation was assessed using two different methods: the incorporation of 5-bromo-2’-deoxyuridine (BrdU, Nacalai Tesque) into DNA and the measurement of DNA concentration.

Briefly, after the incorporation of BrdU into DNA, the cells were fixed and permeabilized. DNA was hydrolyzed using 2M of hydrochloric acid (Nacalai Tesque). After the nuclei were stained with DAPI (Dojindo), the cells were observed under a BZ-X800 integrated fluorescence microscope (Keyence). The percentage of BrdU-positive cells was determined by counting the total number of nuclei in 10 randomly captured images from each well.

MC3T3-E1 cells were plated, the medium was replaced, and the cells were cultured for 24, 72, or 120 h to assess proliferation. The number of viable cells at each time point was determined by measuring the DNA content of the respective samples using PicoGreen^®^ dsDNA Assay Kit (Invitrogen, Thermo Fisher Scientific, Waltham, MA, USA).

### 4.6. Cell Survival Assay

Cell survival was measured using the WST-8 assay. MC3T3-E1 cells were incubated until confluent, and the medium was changed to one with or without EAAs. The number of viable cells was determined by measuring the absorbance of formazan using the Cell Count Reagent SF (Nacalai Tesque).

### 4.7. DNA Damage Detection

MC3T3-E1 cells were cultured in the presence or absence of EAAs for 24 h. DNA damage was detected using the DNA Damage Detection Kit (Dojindo) with an anti-phosphorylated histone H2AX (γH2AX) primary antibody according to the manufacturer’s instructions. The cells were observed under a BZ-X800 all-in-one fluorescence microscope (Keyence), and the average fluorescence intensity of the cells was analyzed using ImageJ (Ver. 1.53e, Wayne Rasband and contributors, National Institutes of Health, USA; http://rsb.info.nih.gov/ij, accessed on 10 December 2021).

### 4.8. Apoptosis Detection

The apoptosis rate of MC3T3-E1 cells was assessed using the annexin V-FITC Apoptosis Detection Kit (Nacalai Tesque). After treatment, the cells were removed using trypsin, washed with phosphate-buffered saline, and resuspended in annexin-binding buffer. Fluorescein isothiocyanate (FITC)-labeled annexin V and PI solutions were added to the cells, which were then incubated for 15 min. The stained cells were observed using a BZ-X800 all-in-one fluorescence microscope (Keyence), and the percentage of apoptotic cells was quantified using flow cytometry (FACSVerse™, BD Biosciences, Franklin Lakes, NJ, USA).

### 4.9. Gene Expression Analysis in MC3T3-E1 Cells

Total cellular RNA was extracted using the RNeasy Mini Kit (Qiagen, Venlo, The Netherlands). The total RNA was reverse transcribed to complementary DNA using a PrimeScript™ reverse transcript kit obtained from Takara Bio (Shiga, Japan). The expression of p53, bax, and gapdh was quantified using TaqMan Gene Expression Analyzer (Thermo Fisher Scientific). Real-time PCR was performed using a QuantStudio^®^ 3 PCR system (Applied Biosystems Real-time PCR systems, Thermo Fisher Scientific).

### 4.10. Western Blot Analysis

MC3T3-E1 cells were treated with or without EAAs for 24, 72, and 120 h and then harvested. The total protein was extracted, normalized, and separated via SDS-PAGE. Then, the samples were incubated with antibodies against p53, Bax, caspase-3, cleaved caspase-3, and β-actin. The membrane was then incubated with a secondary antibody, and the immunoreactive bands were visualized using Chemi-Lumi One L (Nacalai Tesque). Afterward, the primary and secondary antibodies were stripped using a striping buffer and re-probe with other molecular weight proteins on the same membrane. Grayscale values were evaluated using ImageJ.

### 4.11. Statistical Analysis

This study’s data were analyzed and reported as the mean plus standard deviation (SD). Parametric data underwent one-way analysis of variance (ANOVA), and post hoc comparisons were conducted using Tukey’s test. The statistical analysis was carried out using SPSS Statistics Ver. 17 (IBM, Chicago, IL, USA).

## 5. Conclusions

In this study, we investigated the mechanism underlying apoptosis triggered by EAA starvation. Under EAA deprivation, we observed a notable induction of ROS production in MC3T3-E1 cells. This increase in ROS levels caused DNA damage and initiated apoptosis. To validate this observation, we used NAC to block ROS formation. Remarkably, upon blocking ROS formation with NAC, cellular activity was effectively restored. These findings highlight the pivotal role of ROS in mediating the apoptotic response during EAA starvation.

## Figures and Tables

**Figure 1 ijms-24-15314-f001:**
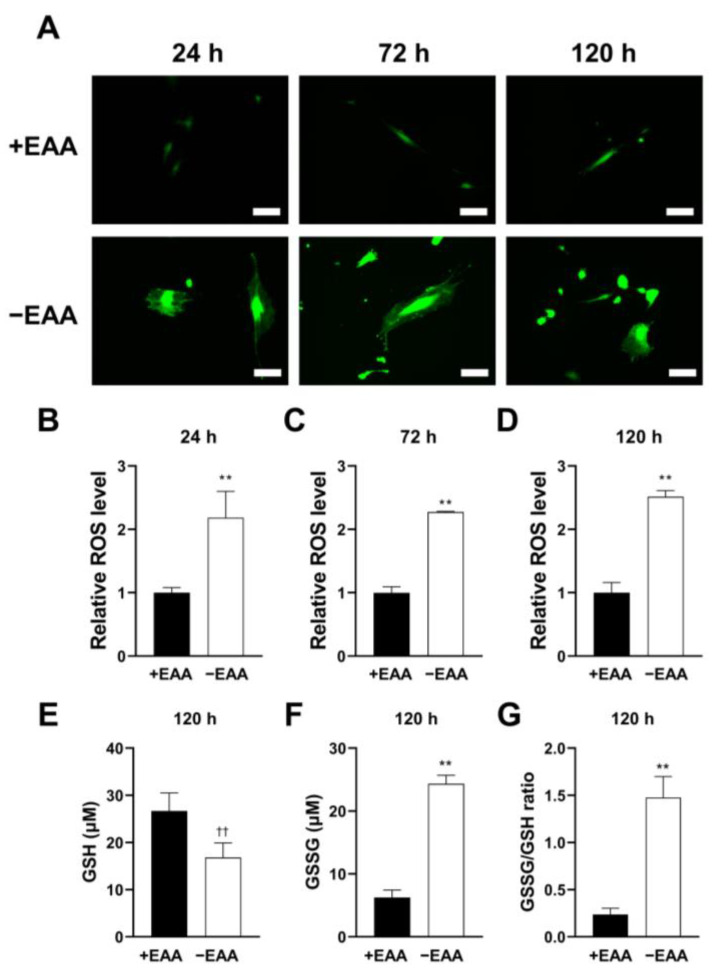
Essential amino acid (EAA) starvation induces reactive oxygen species (ROS) in MC3T3-E1 cells. (**A**) MC3T3-E1 cells stained the total ROS after incubation for 24, 72, and 120 h and were photographed under a fluorescent microscope. (**B**–**D**) Total ROS levels at each time point relative to those in the +EAA group (control group). (**E**–**G**) Levels of reduced glutathione (GSH) and oxidized glutathione (GSSG) and their ratio in stimulated MC3T3-E1 cells incubated for 120 h. Scale bars: 100 μm. A significant increase compared with the control is described by ** *p* < 0.01, †† *p* < 0.01.

**Figure 2 ijms-24-15314-f002:**
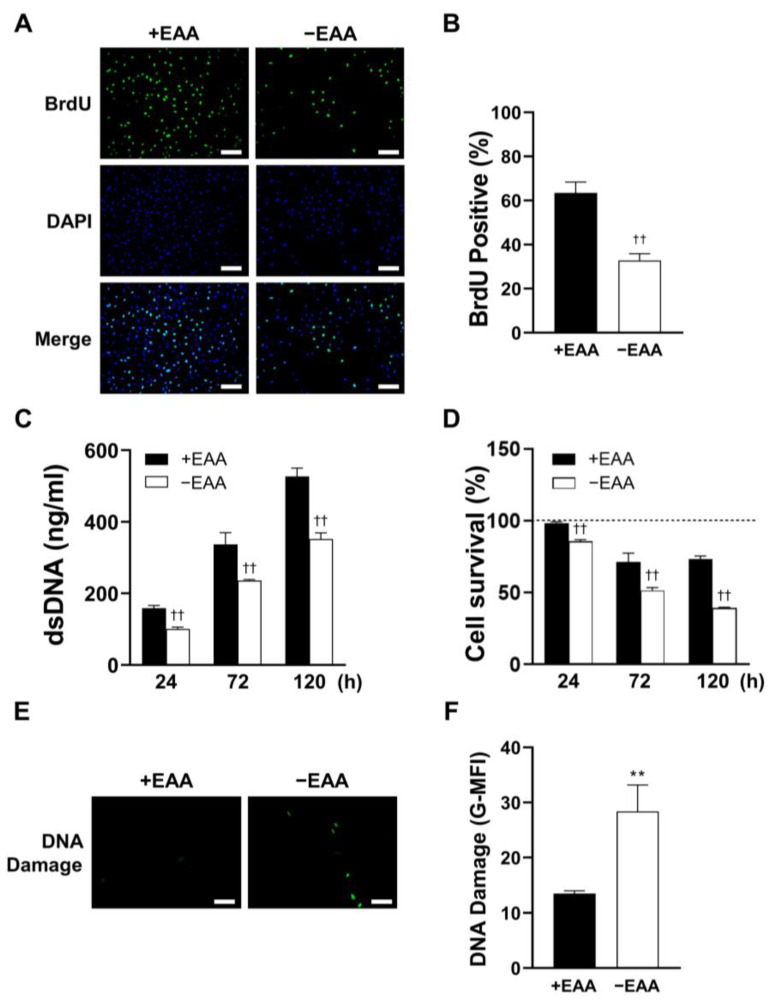
Essential amino acid (EAA) starvation conditions impair cellular viability and induce DNA damage in MC3T3-E1 cells. (**A**,**B**) Representative images of immunofluorescence incorporation of BrdU into DNA and nuclei staining with DAPI. The number of BrdU-positive cells was quantitatively analyzed and is presented as the means ± SD of three independent experiments. (**C**) The total amount of DNA in MC3T3-E1 cells determined after 24, 72, and 120 h of incubation. (**D**) Cell survival rate detected using the MTT assay. For the calculation of relative results, the mean value of absorbance at 0 h for each group was taken as 100%. (**E**,**F**) Representative images of DNA damage after 72 h of incubation of stimulated MC3T3-E1 cells. The mean fluorescence intensity was analyzed using ImageJ Ver. 1.53e. Scale bars: 100 μm. A significant increase compared with the control was described as ** *p* < 0.01. A significant decrease compared with the control was described as †† *p* < 0.01.

**Figure 3 ijms-24-15314-f003:**
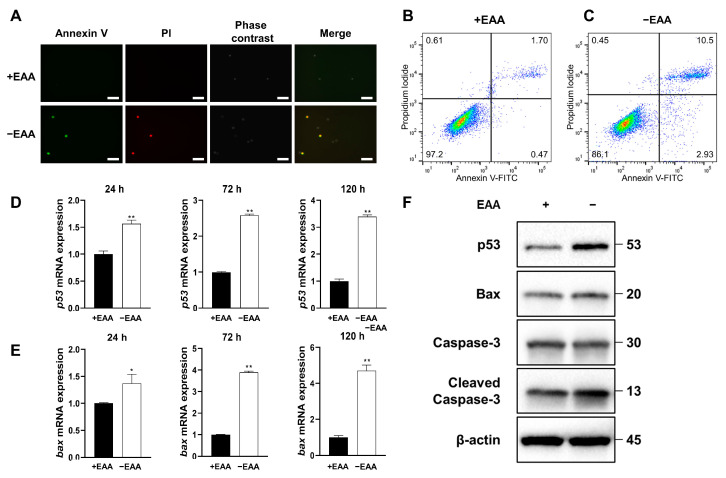
Essential amino acid (EAA) starvation induces apoptosis in MC3T3-E1 cells. (**A**) Representative image of MC3T3-E1 cell apoptosis after 24 h of incubation. Cells stained with both annexin V and PI were considered apoptosis-positive. (**B**,**C**) Representative scatter plots of PI (y-axis) vs. annexin V (x-axis). (**D**–**F**) Expression of p53, Bax, and caspase-3 detected using real-time PCR and Western blotting (Appendix A). Protein extracts from MC3T3-E1 cells were assayed using antibodies against the respective proteins, with β-actin as a loading control. All Western blot samples are from the same experiment, and the gels/blots were processed in parallel. Scale bars: 100 μm. A significant increase compared with the control was described as * *p* < 0.05, ** *p* < 0.01.

**Figure 4 ijms-24-15314-f004:**
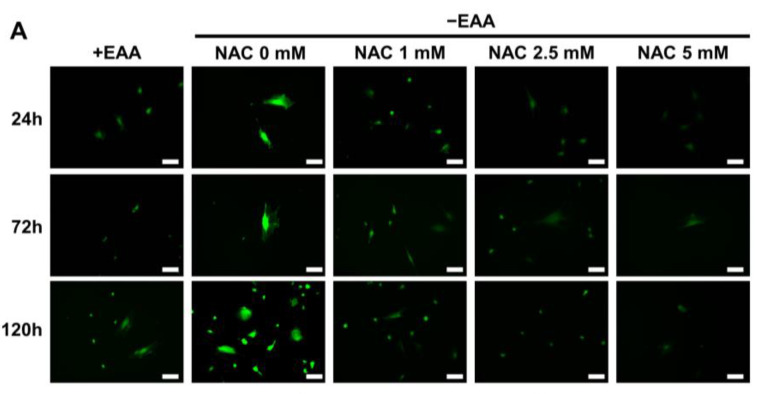
Effect of N-acetyl-L-cysteine (NAC) on reactive oxygen species (ROS) production under essential amino acid (EAA) starvation. (**A**) Representative images of total ROS production in MC3T3-E1 cells stained and photographed after the addition of 1 mM, 2.5 mM, and 5 mM of NAC and incubation for 24, 72, and 120 h. (**B**–**D**) Total ROS levels at each time point with 1 mM, 2.5 mM, and 5 mM of NAC were measured and modified relative to the +EAA group (control group). (**E**–**G**) Levels of reduced glutathione (GSH) and oxidized glutathione (GSSG) and their ratio in stimulated MC3T3-E1 cells after incubation with NAC for 120 h. Scale bars: 100 μm. A significant increase compared with the control was described as ** *p* < 0.01. A significant decrease compared with the −EAA group was described as † *p* < 0.05, †† *p* < 0.01.

**Figure 5 ijms-24-15314-f005:**
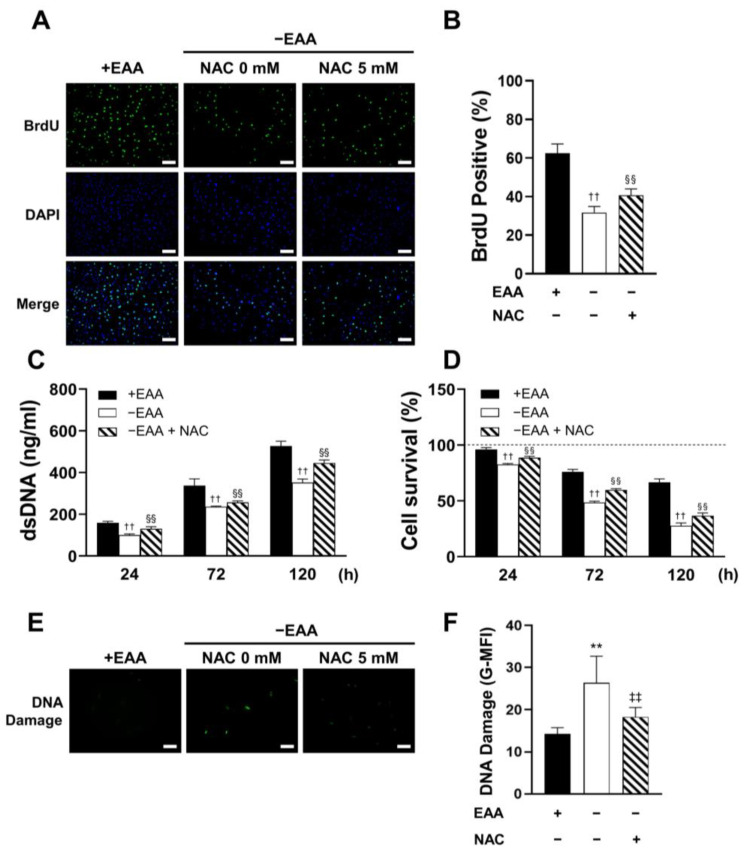
N-acetyl-L-cysteine (NAC) rescues cellular viability and reduces DNA damage in MC3T3-E1 cells under essential amino acid (EAA) starvation. (**A**,**B**) Representative immunofluorescence images of BrdU incorporated into DNA after the administration of 5 mM of NAC. The quantitative analysis of data is presented as the means ± SD, as described before. (**C**) The total amount of DNA in MC3T3-E1 cells after NAC addition was determined after 24, 72, and 120 h of incubation. (**D**) Cell survival was assayed after NAC addition. The relative results are presented as described before. (**E**,**F**) Representative image of DNA damage after 72 h of incubation following NAC addition in MC3T3-E1. The mean fluorescence intensity of the image was analyzed using ImageJ. Scale bars: 100 μm. A significant increase compared with the control was described as ** *p* < 0.01. A significant decrease compared with the group of control was described as †† *p* < 0.01. A significant increase compared with the 0 mg/dL group was described as §§ *p* < 0.01. A significant decrease compared with the group of 0 mg/dL was described as ‡‡ *p* < 0.01.

**Figure 6 ijms-24-15314-f006:**
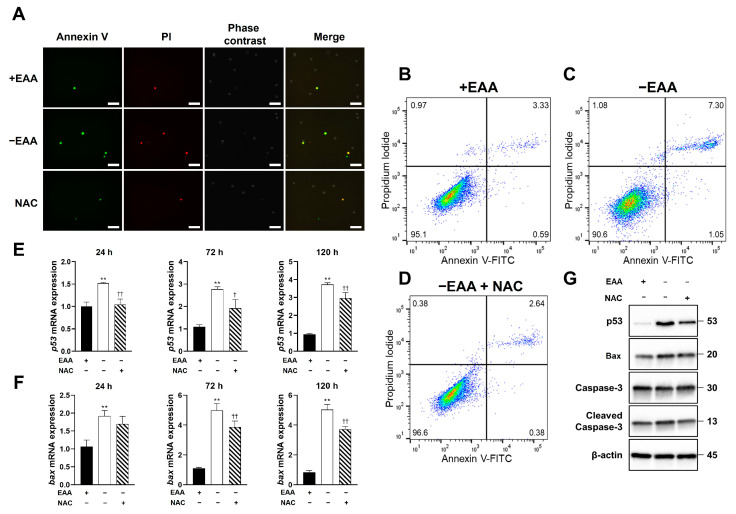
N-acetyl-L-cysteine (NAC) prevents apoptosis induction in MC3T3-E1 cells under essential amino acid (EAA) starvation and decreases the expression of apoptotic genes p53, Bax, and caspase-3. (**A**) Representative image of apoptosis in MC3T3-E1 cells after NAC addition and 24 h of incubation. (**B**–**D**) Representative scatter plots of PI (y-axis) vs. annexin V (x-axis) after NAC administration. (**E**–**G**) Expression of p53, Bax, and caspase-3 detected using real-time PCR and Western blotting (Appendix A). The protein expression was quantified using ImageJ. All samples were from the same experiment, and the gels/blots were processed in parallel. Scale bars: 100 μm. A significant increase compared with the control is described as ** *p* < 0.01. A significant decrease compared with the −EAA group is described as † *p* < 0.05, †† *p* < 0.01.

## Data Availability

The datasets generated during and/or analyzed during the current study are available from the corresponding author on reasonable request.

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
