# Peer review of "Essential Amino Acid Starvation-Induced Oxidative Stress Causes DNA Damage and Apoptosis in Murine Osteoblast-like Cells"

_ijms, 2023, doi:10.3390/ijms242015314_

Round 1

Reviewer 1 Report

In this manuscript the authors explore the mechanisms related to cellular responses to essential amino acids starvation, focusing on ROS-induced DNA damage and apoptosis. The experimental design is basic but appears to be well structured. However the work presents usefulness in terms of knowledge relating to some biochemical aspects, but several points need to be clarified and implemented.

- The work has limits. The conditions dictated by EAA starvation should be reproduced in animal models. This limitation must be explicitly mentioned in the work in the Discussion section.

- I find inappropriate to mention MC3T3-E1 cells in the title. The authors should reformulate the title more generally and explicitly mention essential amino acids (EAAs).

-To reproduce the starvation condition, why did author use MC3T3-E1 cells ?

-is it an immortalized cell line? if yes, describe it.

-line 121: The author mentioned NAC treatment. Explicitly describe the rationale used for the quantities (1, 2.5, and 5 mM). Furthermore, are these quantities clinically compatible?

-Did the authors also evaluate Bcl-2 levels in apoptosis?

- Did the authors evaluate the expression levels of γH2AX by RealTime PCR for the evaluation of DNA damage? If it has not been done, please give reasons.

- Did the authors evaluate the implications on lipid peroxidation caused by the starvation condition of EAAs? For example the most commonly used lipid marker of oxidative stress is Malondialdehyde (MDA).

-The translational purpose of this knowledge in the clinical field is not clear in the work. Authors should specifically explain the clinical applicability of this knowledge.

- In the introduction, mention and describe a pathological and/or clinical condition where NAC can be applied.

Moderate editing of English language required

Author Response

Response to Reviewer 1 Comments

We would like to extend our sincere gratitude to you for your valuable and insightful suggestions, which were greatly beneficial in improving the quality of our article. We have carefully read your comments and taken them into consideration while revising our manuscript. In the following statements, we have provided detailed responses to the comments point-by-point. We hope our responses and the revised article will address comments raised by you. We would like to thank you again for your kind and critical suggestions about our study.

Comments:  In this manuscript the authors explore the mechanisms related to cellular responses to essential amino acids starvation, focusing on ROS-induced DNA damage and apoptosis. The experimental design is basic but appears to be well structured. However the work presents usefulness in terms of knowledge relating to some biochemical aspects, but several points need to be clarified and implemented.

Response: Thank you for your kind comment. Your comments have motivated us to move forward.

Point 1: The work has limits. The conditions dictated by EAA starvation should be reproduced in animal models. This limitation must be explicitly mentioned in the work in the Discussion section.

Response 1: We acknowledge the reviewers' insightful suggestion regarding the reproduction of EAA starvation conditions in animal models. We recognize the importance of such models and have incorporated this limitation into the Discussion section, explicitly highlighting the need for future studies employing animal models to further validate our findings.

Page 10-11 Line 292-298

Moreover, replicating this experiment in an animal model poses challenges due to the absence of dedicated transporters for each amino acid. Blocking the uptake of a specific essential amino acid becomes intricate since amino acids share various transport systems rather than having individualized transporters. To address this complexity, our future studies will aim to attain a similar effect by selectively inhibiting these shared transporters in in vivo experiments.

Point 2: - I find inappropriate to mention MC3T3-E1 cells in the title. The authors should reformulate the title more generally and explicitly mention essential amino acids (EAAs).

Response 2: We appreciate the suggestion to reformulate the title more generally. The revised title now explicitly mentions essential amino acids without specifying a particular cell line.

Page 1 Line 2-4

Essential Amino acid starvation-induced oxidative stress causes DNA damage and apoptosis in MC3T3-E1 cells murine osteoblast-like cells

Point 3: To reproduce the starvation condition, why did author use MC3T3-E1 cells ?

Response 3: The use of MC3T3-E1 cells to reproduce EAA starvation conditions was based on our antecedent studies exploring the ability of EAA starvation for osteoblast differentiation and its well-established application in osteoblast research. We acknowledge that the rationale behind this choice could be better articulated, and we have added a brief explanation in the discussion section.

Page 8 Line 187-190

MC3T3-E1 is a mouse osteoblast cell line commonly used in research related to bone biology and osteogenesis, which are frequently utilized in vitro to study various aspects of bone biology, including osteoblast differentiation, mineralization, and response to different factors.

Point 4: is it an immortalized cell line? if yes, describe it.

Response 4: Yes, MC3T3-E1 is an immortalized cell line. We have now included a brief description of MC3T3-E1 cells in the Materials and Methods section.

Page 11 Line 301

MC3T3-E1 cells, an immortalized cell line, were purchased from RIKEN BioResource Research Center (Tsukuba, Ibaraki, Japan) and cultured in Dulbecco’s Modified Eagle’s Medium (Nacalai Tesque, Kyoto, Japan) with or without the nine EAAs and supplement-ed with 10% fetal bovine serum (Hyclone, Thermo Fisher Scientific, Waltham, MA, USA).

Point 5: line 121: The author mentioned NAC treatment. Explicitly describe the rationale used for the quantities (1, 2.5, and 5 mM). Furthermore, are these quantities clinically compatible?

Response 4: We appreciate the reviewers' request for clarification on the rationale behind NAC treatment quantities. The effective concentration of NAC exhibits variability across different cell types. To establish our experimental parameters, we referred to and adopted concentrations that have been employed in prior studies [1,2]. This approach is in alignment with the established practices within the scientific community, allowing for a comparative framework and ensuring consistency with existing literature.

   Furthermore, the dosage and concentration of NAC are subject to variation based on the specific medical indication. In a clinical context, a standard intravenous loading dose might be around 150 mg/kg administered over 15 minutes, followed by a maintenance dose. For instance, in a human weighing 80 kg, this regimen could result in a mean plasma concentration of approximately 15.25 mM, while the maximum human dose is 600-1000 mg/kg [3]. The concentrations employed in our study did not surpass these clinical levels, aligning with safety practices and ensuring that our experimental conditions were within clinically relevant bounds.

Point 6: Did the authors also evaluate Bcl-2 levels in apoptosis?

Response 6: We appreciate the reviewer's insightful suggestion regarding the investigation of Bcl-2 levels in apoptosis. Unfortunately, due to time constraints in responding to the review comments, obtaining the necessary reagents for real-time PCR analysis of Bcl-2 levels has proven challenging. Recognizing the importance of probing Bcl-2 as a key regulator of apoptosis, we acknowledge that our study intentionally focused on upstream events (p53 and Bax) and downstream events (caspase-3). The emphasis was on a comprehensive analysis of the apoptotic pathway, with attention to these key elements. While Bcl-2 was not included in our current analyses, we value the reviewer's input and are committed to considering this aspect in future investigations when practical.

Point 7: Did the authors evaluate the expression levels of γH2AX by RealTime PCR for the evaluation of DNA damage? If it has not been done, please give reasons.

Response 7: Thank you for your kind comment. The decision not to assess γH2AX expression through real-time PCR aligns with the rationale that phosphorylation of histone H2AX, a crucial step in DNA damage response, specifically occurs in response to double-strand breaks (DSBs). This phosphorylation is orchestrated by kinases in the PI3K pathway. γ-H2AX serves as an early marker, initiating the recruitment and localization of DNA repair proteins. The rapid formation of γ-H2AX foci in response to DSBs makes them a reliable biomarker of damage, presenting a 1:1 correlation with the occurrence of breaks.

Given that antibodies against γ-H2AX are available, immunofluorescence provides a direct means of detecting these foci using secondary antibodies. In contrast, real-time PCR values are more suited for measuring the expression of H2AX, making immunostaining at this juncture a more representative approach for capturing and quantifying DNA damage. This methodological choice enhances the precision of our assessment of DNA damage in the context of the experiment.

Point 8: Did the authors evaluate the implications on lipid peroxidation caused by the starvation condition of EAAs? For example the most commonly used lipid marker of oxidative stress is Malondialdehyde (MDA).

Response 8: Unfortunately, we did not evaluate lipid peroxidation in response to EAA starvation. We acknowledge this limitation and have addressed it in the Limitations section. Future studies should consider assessing lipid peroxidation using markers such as Malondialdehyde (MDA).

Page 10 Line 281-288

However, a major limitation of this study is that we did not examine the individual roles of each EAA in mediating oxidative stress and apoptosis. Furthermore, the study did not explore other potential oxidative responses resulting from the deprivation of EAAs, such as the effect on Malondialdehyde (MDA) in lipid peroxidation reactions. Investigating these additional facets of oxidative stress could offer a more comprehensive understanding of the broader effects induced by EAA starvation. Recognizing these limitations, future research endeavors may benefit from a more detailed exploration of the individual roles of specific EAAs and an expanded examination of oxidative responses beyond the scope of DNA damage and apoptosis.

Point 9: The translational purpose of this knowledge in the clinical field is not clear in the work. Authors should specifically explain the clinical applicability of this knowledge.

Response 9: We recognize the importance of highlighting the translational implications of our findings. The revised Discussion section now explicitly discusses the clinical applicability of our knowledge, emphasizing potential implications for conditions involving oxidative stress.

Page 10 Line 257-267

Furthermore, the ability of NAC to rescue cellular damage induced by ROS highlights its potential therapeutic significance in countering apoptosis under EAA deficiency. Moreover, the efficacy of NAC in ameliorating cellular damage induced by ROS underscores its potential therapeutic significance in mitigating apoptosis resulting from EAA deficiency. This discovery not only highlights the central role of ROS in orchestrating the observed cellular responses but also positions NAC as a promising candidate for therapeutic intervention to counteract the deleterious effects of apoptosis in the context of EAA deficiency. By specifically targeting ROS-mediated DNA damage and apoptosis, NAC emerges as a compelling candidate for therapeutic strategies aimed at mitigating the ad-verse cellular effects associated with EAA deficiency. This has far-reaching implications for the development of targeted interventions to enhance cellular resilience and overall clinical outcomes in situations of EAA deficiency.

Point 10: In the introduction, mention and describe a pathological and/or clinical condition where NAC can be applied.

Response 10: We appreciate the suggestion to mention a pathological or clinical condition where NAC can be applied in the Introduction. The revised Introduction now includes a brief discussion on the therapeutic role of NAC.

Page 2 Line 61-67

In this study, we employed N-Acetylcysteine (NAC), a versatile drug and supplement with various clinical applications, to eliminate the effects of ROS. NAC's uses range from serving as an antidote to supplementing cellular glutathione oxidants and treating specific psychiatric disorders.  In this study, we sought The findings of this study are anticipated to significantly advance our understanding of how cells respond to EAA deficiencies, seeking to bridge the existing knowledge gap surrounding cellular responses to EAA deficiency and gain insights into the molecular pathways implicated in these cellular changes.

We believe these revisions substantially improve the manuscript's quality and appreciate the invaluable input provided by the reviewers. We hope these changes address the concerns raised and contribute to the overall merit of our work.

Reference:

  1. Liu, M.; Wu, X.; Cui, Y.; Liu, P.; Xiao, B.; Zhang, X.; Zhang, J.; Sun, Z.; Song, M.; Shao, B. Mitophagy and Apoptosis Mediated by ROS Participate in AlCl3-Induced MC3T3-E1 Cell Dysfunction. Food Chem. Toxicol. 2021, 155, 112388.
  2. Lee, D.; Kook, S.-H.; Ji, H.; Lee, S.-A.; Choi, K.-C.; Lee, K.-Y.; Lee, J.-C. N-Acetyl Cysteine Inhibits H2O2-Mediated Reduction in the Mineralization of MC3T3-E1 Cells by down-Regulating Nrf2/HO-1 Pathway. BMB Rep. 2015, 48, 636.
  3. Nolin, T.D.; Ouseph, R.; Himmelfarb, J.; McMenamin, M.E.; Ward, R.A. Multiple-Dose Pharmacokinetics and Pharmacodynamics of N-Acetylcysteine in Patients with End-Stage Renal Disease. Clin. J. Am. Soc. Nephrol. 2010, 5, 1588–1594, doi:10.2215/CJN.00210110.

Reviewer 2 Report

This is an interesting study where the research group investigated the impact of EAA starvation in MC3T3-E1 cells. They delineated the mechanism through the finding that EAA starvation-induced generation of reactive oxygen species (ROS) resulted in apoptosis and DNA damage in the above mentioned cells. As a whole, the study is novel, data are clearly presented, and English language is perfect. However, to further improve the findings, I have the following suggestions:

a) Please incorporate the loading control (Beta-actin) figures for individual protein of interest. This is applicable to Fig. 3 and 6. I understand that the researchers included the original, unedited western blot data. Nevertheless, unless the same membrane/blot was used to re-probe the proteins, we cannot confirm the equal loading of the protein lysates.

Author Response

Response to Reviewer 2 Comments

We would like to extend our sincere gratitude to you for your valuable and insightful suggestions, which were greatly beneficial in improving the quality of our article. We have carefully read your comments and taken them into consideration while revising our manuscript. In the following statements, we have provided detailed responses to the comments point-by-point. We hope our responses and the revised article will address comments raised by you. We would like to thank you again for your kind and critical suggestions about our study.

Comments and Suggestions for Authors: This is an interesting study where the research group investigated the impact of EAA starvation in MC3T3-E1 cells. They delineated the mechanism through the finding that EAA starvation-induced generation of reactive oxygen species (ROS) resulted in apoptosis and DNA damage in the above mentioned cells. As a whole, the study is novel, data are clearly presented, and English language is perfect. However, to further improve the findings, I have the following suggestions:

Response: Thank you for your kind comment. Your comments have motivated us to move forward.

Point 1: Please incorporate the loading control (Beta-actin) figures for individual protein of interest. This is applicable to Fig. 3 and 6. I understand that the researchers included the original, unedited western blot data. Nevertheless, unless the same membrane/blot was used to re-probe the proteins, we cannot confirm the equal loading of the protein lysates.

Response 1: We acknowledge the importance of ensuring the equal loading of protein lysates, and we would like to clarify our experimental approach. In our study, we utilized striping buffer to remove antibodies from the membrane, allowing us to re-probe the same membrane for other proteins with different molecular weight positions, including the loading control β-actin. Consequently, the protein lysates are deemed identical across all lanes. This approach has been explicitly detailed in the revised materials and methods section. Furthermore, we confirm that all membranes utilized in this study underwent re-probing specifically for β-actin and uploaded with the letter.

Page 12 Line 342-343

The membrane was then incubated with a secondary antibody, and the immunoreactive bands were visualized using Chemi-Lumi One L (Nacalai Tesque). After this, the primary and secondary antibodies were stripped using striping buffer and re-probe other molecular weight proteins on the same membrane.

Below is the re-probed β-actin.

We believe that these clarifications address the concern raised by the reviewer and strengthen the reliability of our findings. We appreciate the opportunity to improve the manuscript based on these valuable comments.
